# Inequalities in Health: Methodological Approaches to Spatial Differentiation

**DOI:** 10.3390/ijerph182312275

**Published:** 2021-11-23

**Authors:** Dana Hübelová, Martina Kuncová, Hana Vojáčková, Jitka Coufalová, Alice Kozumplíková, Francois Stefanus Lategan, Beatrice-Elena Chromková Manea

**Affiliations:** 1Department of Social Studies, Faculty of Regional Development and International Studies, Mendel University in Brno, 613 00 Brno, Czech Republic; beatrice-elena.manea@mendelu.cz; 2Department of Economic Studies, College of Polytechnics Jihlava, 586 01 Jihlava, Czech Republic; kuncova@vspj.cz; 3Department of Technical Studies, College of Polytechnics Jihlava, 586 01 Jihlava, Czech Republic; vojackova@vspj.cz; 4Department of Development, City Municipality of Břeclav, 690 02 Břeclav, Czech Republic; jitka.coufalova@breclav.eu; 5Department of Environmental Studies, Faculty of Regional Development and International Studies, Mendel University in Brno, 613 00 Brno, Czech Republic; 6Department of Regional and Business Economics, Faculty of Regional Development and International Studies, Mendel University in Brno, 613 00 Brno, Czech Republic; francois.lategan@mendelu.cz

**Keywords:** inequalities in health, health determinants and indicators, composite indicator, districts of the Czech Republic

## Abstract

The prevalence of inequalities in the general health position of communities can be assessed by using selected determinants. The aims of this article are three-fold: (1) to apply a comprehensive approach to the assessment of inequalities in the general health position of communities, (2) to determine the spatial differentiation of determinants, and (3) to present selected assessment methods and their impact on the results. To present a quantitative assessment of these inequalities in health status in communities, a composite indicator (Health Index) was developed. This Health Index is composed of 8 areas of evaluation and 60 indicators which include, amongst others, determinants of health status and healthcare at district level (LAU 1) in the Czech Republic. The data are evaluated using multicriteria decision-making methods (the WSA and TOPSIS methods). Findings suggest that, when all eight domains are assigned the same weight of one, the spatial differentiation among the districts is similar when using both methods. If different weightings are assigned to the districts, changes occur in both the index values and the rankings of the analyzed districts. For example, the allocation of weightings in both methods results in a rearrangement of the ranking of districts for which the Health Index is around the average.

## 1. Introduction

Inequalities in the general health position of communities are understood as unequal differences resulting from inequalities in determinants belonging to the social, economic, or environmental domain. Ideally, all people should have an equal opportunity to reach his or her full health potential. Health equity means that everyone has equal opportunity to be as healthy as possible [1]. This requires the removal of important barriers to a good health position for communities such as poverty, discrimination (including a lack of access to good jobs with fair pay), quality education and housing, safe environment, and healthcare [2]. Searching for the reasons for such inequalities is a very complex process, as it hides a number of linkages both between indicators and in their overarching areas [3]. Efforts to measure inequalities in the general health position of communities are not new [4], but the definition of impartiality and (in)equality and the methodology to measure them, including the identification of optimal indicators, are controversial [5]. In [6] ‘social model of health’, health inequalities are commonly analyzed along several dimensions relating to social, economic, and environmental determinants. In order to better understand why some populations or groups are healthier than others, and to take actions that will improve health and reduce inequalities, the monitoring of determinants of health should go beyond measuring health outcomes [7]. For this reason, a shift in monitoring determinants and indicators, including causes and risk factors, is needed [8,9]. In particular, the task of monitoring population health inequalities using multidimensional indicators requires the availability of spatially disaggregated data.

Objective identification and monitoring of health inequalities is essential on two levels: (1) to improve the average quality of health in the population and (2) to reduce inequalities in achieving good health conditions themselves [10]. The creation of a quality and sustainable environment, and an adequate level of economic and social development, simultaneously promotes good health conditions and social equity [11]. In this paper, we aim to reflect on the above issues, initially presenting preliminary results aimed at developing a substantive classification and subsequent quantification of the complex of relevant determinants of health. The health index in the Czech population and its spatial differentiation will be presented to illustrate the methodological approach. A composite indicator—the Health Index—was created to assess health inequalities. The index is composed of eight areas, which make up 60 indicators, which are analyzed in 77 regions of the Czech Republic. In comparing the results according to the mathematics methods, very similar tendencies can be observed. The paper presents a holistic approach to the assessment of determinants affecting health conditions at both the individual and population levels. This approach also enables the identification of problems whose solution can lead to the elimination or at least justification of health inequalities.

## 2. Determinants of Health and Measurement of Inequalities in Health Conditions

The choice of determinants in the measurement of inequalities in health conditions in communities is directed by the fact that health is influenced by a compound of interactions between all factors belonging to the “life space” [6,12]. These are interconnected and evolve over time and space [13]. We consider the components of “life space” to be the external environment (water, air, and soil), living conditions (individual environmental quality, access to healthcare, etc.), and socio-economic conditions (economic performance, socio-economic status, education, social disadvantage, etc.) [14]. Conceptual issues of spatial differences in health indicators and inequalities have been addressed, for example, in studies [15,16,17,18]. In recent decades, the possibilities of using GIS in the analysis of spatial inequalities have been intensively discussed, e.g., [19,20,21].

Determinants of health conditions also influence so-called health outcomes, with the most commonly considered measurable health ‘outcome’ being the mortality structure indicator [22]. Differences in mortality structure indicator values by socioeconomic level are universal [23], regardless of whether objective indicators of health conditions (illness, disability, mortality) or subjective indicators of health conditions (where respondents assess their own health status) are used. The observed trend is always the same, with poorer health condition status observed at lower socioeconomic levels [24]. 

The relationship between health condition determinants and health outcomes was confirmed by a recent study carried out in the United States [25]. Using a composite measurement of determinants of health condition and standardized measures of mortality by cause of death, the index identifies changes and variations across US states. The results indicate that since the 1980s, better values of the index have been found in the wealthiest states (California, Washington DC, and Massachusetts) due to, among other things, improvements in quality of life and healthcare. The results show a steady increase in overall health condition inequality among US states. This inequality is explained by race segregation, ethnicity, migration history, and other stigmatized social statuses [25]. Hacker and colleagues [26] acknowledge the importance of education, housing conditions, and access to good quality healthcare as some of the most important social determinants of health status. These factors along with employment status, job opportunities, and income may either improve or limit individual health status (e.g., [27,28,29]). Other authors emphasize the importance of overall and health literacy (e.g., [30]) or environmental factors (e.g., [31]) in the assessment of health determinants and inequalities. Spatial differentials in health condition are also observed across regions and countries in Europe. Beyond education, the objective cause of spatial differentials in health condition outcomes between countries is also represented by different levels of socioeconomic development [32,33,34,35]. Socio-economic conditions influence the structure of mortality rates in the following way: countries with favorable socio-economic conditions record lower mortality rates (e.g., Finland, Sweden, and France), while countries with worse socio-economic conditions show higher mortality rates (e.g., Bulgaria, Romania, Croatia, Hungary, and Latvia) [36]. Similar results are found in a study investigating the countries in the central and eastern Europe (CEE) region [34]. Results from this study suggest a much higher mortality for CEE countries than countries in other regions of Europe, both for those with low levels of education and those with high levels of education. For Europe as a whole, while it is true that individuals with low levels of education die earlier than those with higher levels of education, there is still a significant difference in mortality rates between eastern and western European countries of the same educational categories, although the trend over the last 30 years indicates that European countries successfully fight against health inequalities [34]. According to the European Core Health Indicators, the life expectancy at birth of the population with a high educational level for CEE countries is similar to the life expectancy at birth of the population with a low educational level in northern Europe or Italy [37].

The above-mentioned disparities also arise within countries (e.g., in the USA according to [38] and in European countries according to [1]). In the Euro-Healthy project, a Population Health Index (PHI) was developed for EU countries at the NUTS2 level. The results show that systematic spatial inequalities persist in Europe at the NUTS2 level. These inequalities are also present within EU countries [33]. All dimensions of socio-economic status (achieved education, employment, economic resources such as income, etc.) influence the health status in the population and result in social inequalities in health conditions, e.g., [39,40,41]. The level of educational attainment is associated with other determinants of health status, from lifestyle-related risk factors to access to preventative programs, health literacy, or the choice of the most appropriate healthcare. 

Despite the plethora of published spatial analyses that track population health and health status, including social and economic phenomena [34,42], etc., there are still methodological questions about how best to measure health inequalities to accurately reflect health affected by environmental, economic, social, cultural, individual factors, healthcare resources, etc. [43]. To grasp health condition in all its complexity [6,12], it is proving desirable to develop metrics that take into account not only the health indicators themselves, but also other dimensions and determinants of health [7,44].

One way to work with the concept of health condition is through the application of composite indicators or indices reflecting (in)equality in health condition [5,25,45,46,47]. On one hand, the method of composite indicators is a recognized technique that allows the explanation of more complex research problems; on the other hand, the optimal methodological constructs and mathematical aspects remain rather unclear [48,49]. At the same time, there is no agreed standard or internationally accepted rule for determining the number and type of variables to be included in health quantification [5,50]. However, indices are a valued, recognized, and useful measurement tool not only for etiological approaches, but also for development strategies and policy decisions, as well as for institutional public communication [51,52]. A review study [5] presents a summary of the literature (more than 1500 studies) on indices for assessing spatial data on the determinants of health and environment. The authors identified 23 indices and assessed 329 variables. This diversity illustrates the lack of a common framework, which can lead to strong subjectivity and at the same time, limit the possibility of comparing in time and across countries or regions different indices of health and environment. The specificity and varying availability of variables also limit the transferability of the indices [5].

The aim of this article is to present the results of a comprehensive approach to the assessment of inequalities in health and their spatial differentiation and to show the selected methods of assessment and their influence on the results. In our research, we contribute to the three levels of the aforementioned problems of health inequalities assessment by: (1) presenting the results of measuring inequalities by selecting a wide range of dimensions and their relevant determinants (e.g., [32]); (2) in contrast to the aforementioned research, assessing inequalities at the (micro-)regional district level (LAU 1), which allows not only for more detailed spatial analyses but also for the subsequent development of specifically targeted proposals for actions leading to the elimination or at least the reduction of health inequalities; and (3) using, validating, and comparing the results of methods integrating technical elements of multi-criteria modeling and data visualization, and social elements of interdisciplinary processes in the form of experts’ opinions on determinants relevant for health assessment (e.g., [53]).

## 3. Materials and Methods

A composite indicator—the Health Index—was developed to assess health condition inequalities. The Health Index includes both determinants and indicators of health (see [54]). The selection of indicators was directed by the following issues: (a) monitoring and availability of data in public databases over time and at the required geographical level (76 districts—LAU 1 and the territory of the Capital City of Praha, a total of 77 regions of the Czech Republic) and (b) the range of available indicators (determinants of health, health status, healthcare, etc.; [55]). The index is composed of eight areas, which make up 60 indicators. The original set of 68 indicators was reduced after correlation analysis to eight indicators that showed a higher correlation value than required, i.e., more than −0.8 or 0.8. Areas 1 to 7 represent determinants of health condition and Area 8 health indicators (or health outcomes). The data come from publicly available databases of the Czech Statistical Office, the Institute of Health Information and Statistics of the Czech Republic, the Ministry of Labor and Social Affairs of the Czech Republic, the Czech Hydrometeorological Institute, and the Czech Household Panel Survey. The data are from 2016, 2017, 2018, and 2019. The only indicators in relation to the level of education and housing conditions come from the 2011 Census of Population, Houses, and Flats.

The newly developed Health Index is a mathematical combination of variables that reflect multiple selected variables [56,57]. Two methods of multicriteria variance scoring were used for calculating it: the WSA and the TOPSIS method. The Weighted Sum Approach (WSA) method is a method based on the principle of utility maximization. The method is based on the assumptions of linearity and maximization of all partial utility functions, which are obtained by normalizing the original input data. The TOPSIS (Technique for Order Preference by Similarity to an Ideal Solution) method aims to select a compromise option with the assumption that the best option has the smallest distance from the ideal option and the largest from the baseline option. In contrast to the WSA method, TOPSIS is based on the principle of minimizing the distance from the ideal variant or maximizing the distance from the basal variant. A more detailed explanation of the used calculation methods is given in the Appendix A. The higher the Health Index value, the more favorable the situation in the region (this is valid for both methods). The calculations were performed in MS Excel Microsoft Corporation, Redmond, DC, USA. and its add-in SANNA 2014 (Josef Jablonský, 2009, Czech Republic) [58].

The Health Index was processed by the WSA and TOPSIS method in two variants: (a) each of the eight areas has equal importance—weight 1 and (b) each of the eight areas has different importance—weightings (the sum of the weightings is 80; see Table 1).

The importance of each area and attached weights were determined by an interdisciplinary expert assessment using the Delphi method [53]. The most used methods to identify the weights were statistical, multi-attribute modeling, and subjective. We worked with subjective methods to determine the weights as we aimed to capture what is more important in the phenomenon we study. These new methods have been introduced with the aim to involve more individuals—experts or citizens—in the process of defining weights systems for social indicators. We chose the Delphi method among the subjective methods for weighting [59,60].

The Delphi methodology is a well-known technique, which relies on a panel of experts. The Delphi method was used because we were interested in collecting opinions from experts in the field of health and health inequalities. Building good quality health indicators requires validity, reliability, and sensitivity. The Delphi technique has high face validity, which is a prerequisite for any quality indicator [61]. This method allowed us to reduce the influence of participants upon one another when determining the weights. Our DELPHI questionnaire was distributed by email in order to maintain confidentiality and avoid leading effects. In our study, we selected a wide range of experts (10 people) from different disciplines (e.g., physicians, sociologists, demographers, economists, statisticians, public health specialist, and others) that come together in this research. Ten independent experts from different disciplines related to population health anonymously determined the scores of each of the eight areas through a questionnaire. The experts’ opinions were refined in a three-round evaluation process and further used to create the Health Index.

## 4. Results

In the following section, we discuss the results, which we have thematically divided into three areas, each presenting a comparison of the WSA and TOPSIS methods, namely: (a) spatial differentiation of health inequality indicators, (b) comparison of benefits using identical and different weightings, and (c) assessment of key determinants and indicators of health.

### 4.1. Spatial Differentiation of Health Inequality Indicators

The topic of spatial differentiation of health inequalities in the districts of the Czech Republic was assessed through the Health Index values. We simultaneously compared the results and differences of two multi-criteria assessment methods (WSA and TOPSIS) when using similar and different area weightings. Spatial differentiations of health inequalities were defined based on the achieved Health Index values.

The highest Health Index values, using both the WSA and TOPSIS methods, were calculated for districts that are close to the regional cities (Praha-západ, Praha-východ) or are directly formed around regional cities (Capital City of Praha, Brno-město, and Plzeň-město). In contrast, the Health Index values are low especially in the districts situated at the Czech western border (Most, Teplice, Louny) and in northern Moravia (Jeseník, Bruntál, Ostrava-město, Karviná). In the case of the TOPSIS method, areas on the border in southern Moravia (e.g, Hodonín) join the above-mentioned districts with the lowest Health Index values (see Table 2; Figure 1).

Regions with a high Health Index are characterized by positive regional disparities (low housing subsidies, low unemployment, high ratio of university students, positive migration balance, etc.). Regions with a low Health Index are characterized by negative regional disparities (especially high housing subsidies, high unemployment, low ratio of university students, negative migration balance, high infant mortality, etc.; [62]). Immigrants from outside Europe play no role here, but there are socially excluded localities with ethnic minorities (Roma) and an increased proportion of people with lower socio-economic status.

When different weightings were assigned to the districts, the Health Index values and the rankings of the districts changed, although the initial spatial differentiation of health condition inequalities remained almost unchanged, as in the case when equal weightings were used for all districts (Table 3). A deterioration of the results was observed especially in the border districts of South Moravia (Figure 2).

### 4.2. Comparison of the Benefits of Using Identical and Different Weightings

For each district, we compared the change in rankings and Health Index values that resulted from changing the weightings for each district when applying both the WSA and TOPSIS methods. Both methods yielded similar results when clustering the districts into groups that are typically characterized by similar combinations of assessed contributions (Figure 3). 

When using the WSA method in calculating the Health Index value, differences in rankings and utility in weighting assignment for districts of the Czech Republic are reflected in Figure 3 (presenting only the results for groups 1, 4, 5, and 6). The findings clearly suggest that the districts in group 1 are characterized by the highest growth in the Health Index and at the same time the highest positive change in the ranking (especially the Capital City of Praha, Mladá Boleslav, Jičín). In the case of group 2, the index value increased, but the ranking did not change significantly. In the case of group 3, although the Health Index value increased, the final ranking decreased when using the same methods with different weightings. All districts in these three groups experienced a growth in the Health Index value. The other districts also form three specific groups, which share a decreasing trend in the Health Index value. In the case of group 4, there was a decrease in the Health Index value without a decrease in the ranking (the most significant decrease in the index out of all districts was in Most, Karviná and Ostrava-město). In the case of group 5, both the Health Index value and the ranking decreased. In the case of group 6, the decrease in the Health Index value was reflected in the most significant fall in the rankings of these districts (e.g., Děčín and Ústí nad Labem).

Using the TOPSIS method of analysis, groups of districts with typical patterns of change in benefit appeared, while some districts changed positions within these groups. Two main differences were observed when comparing the TOPSIS method to the WSA method: (a) the allocation of weightings did not produce such a wide range of differences in benefits, but (b) a more pronounced change in the Health Index values, and rankings affected more districts, either in positive or negative benefits (Figure 4).

Figure 4 reflects the results of using the TOPSIS method of analysis (we present only the results for groups 1, 4, 5, and 6). In the case of group 1, districts with the highest increase in the Health Index value are indicated, showing the highest positive change in ranking is larger. In the case of group 2, the increase in the Health Index value shows a different degree of change in the ranking of districts. In the case of group 3, there was an improvement in the Health Index value and a simultaneous decrease in rankings for a very small number of districts. The other three groups of districts (4 to 6) showed a decrease in the Health Index value and are characterized by a decrease in the Health Index value without a decrease in rankings (group 4). In the case of group 5, the Health Index value decreased and with it the ranking, while in group 6, the decrease in the Health Index value also reflects in the most significant decrease in the ranking of districts (Děčín and Ústí nad Labem). Despite this similarity in the characteristics of the groups of districts when comparing the two methods, the TOPSIS method generally results in a lower negative reduction in the Health Index than the WSA method.

### 4.3. Assessment of the Key Determinants and Indicators of the Health Position of Communities

The third research area focuses on identifying the key determinants and indicators of health positions that underlie health inequalities. We analyzed those districts from which we deliberately selected districts belonging to the same groups, represented by the highest and lowest Health Index values calculated by both the WSA and TOPSIS methods with weightings (see Table 3; Figure 2). To identify the key determinants and indicators (outcomes) of the health condition of communities that also reflect both positive and negative inequalities, we used a decomposition of the Health Index into areas (Table 4, Figure 3) and, within areas, into sub-indicators.

We first present the decomposition as a comparison of the results of all areas by districts having a high WSA Health Index value calculated with weightings (Praha-západ, Praha-východ, Brno-město, České Budějovice, and Capital city of Praha). The main contributors to the positive scores for these districts are Area 2 (education; weight 0.18) and Area 8 (health; weight 0.20). In Area 2, education is characterized by an above-average proportion of people with a university degree and a low proportion of people with incomplete or primary education. In Area 8, health conditions are associated with above-average life expectancy and below-average total standardized mortality, as well as below-average mortality rate by underlying causes of death, including deaths caused by tobacco smoking and diabetes mellitus. With the exception of the district of Brno-město, the results are also favorable in Area 1 (economic conditions and social protection; weight 0.19) and Area 6 (except for the Capital City of Praha) (road safety and crime; weight 0.04). In Area 1, the value of the economic conditions and social protection index is reduced by the above average unemployment rates of both gender and age in the case of the Brno-City district. In the Capital City of Praha, the negative result in Area 6 regarding the road safety and crime index is directed by an above-average share of traffic accidents and registered crimes. In Area 3 (demographic indicators; weight 0.08), most districts are downgraded, mainly due to the higher age index. However, the districts of Praha-východ and Praha-západ remain in a favorable position. Area 5 (individual living conditions; weight 0.09) shows a similar regional distribution. On the other hand, in Area 7 (health and social care resources; weight 0.10), the districts of Praha-východ and Praha-západ lag significantly behind the other districts due to negative results for the Hospital Bed Capacity indicator. Area 4 (environmental conditions; weight 0.14) is rather unfavorable for this group of districts, with above average values of air pollution and a low coefficient of ecological stability. The only exception is the České Budějovice district (see Table 4).

The same procedures of decomposition and comparison of the results of these areas were also performed for districts with a low Health Index (Jeseník, Louny, Sokolov, Teplice, Bruntál, Chomutov, Ostrava-City, Most, and Karviná), again based on the WSA method with weightings. This group of districts is mainly associated with unfavorable results in Area 1 (economic conditions and social protection; weight 0.19), Area 8 (health; weight 0.20), and Area 2 (education; weight 0.18) except for Ostrava-City district. Negative values also prevail in Area 6 (road safety and crime; weight 0.04), where only the district of Karviná fares slightly better. The other areas 3, 4, 5, and 7 could not be clearly assessed within this group of districts. In Area 3 (demographic indicators; weight 0.08), the index values widely range from the highest (urbanized districts, Ostrava-City, Most, Chomutov, Sokolov) with a rather younger or average age structure, to a low index in the district Jeseník, which is characterized by an above-average age index and a migration loss. In terms of the extent of the differences in scores, the situation is also very similar in Area 4 (environmental conditions; weight 0.14): above-average index in the districts of Sokolov, Jeseník, Chomutov, and Bruntál and lowest in the districts of Most, Karviná, and Ostrava-City. The results are also inconsistent in Area 5 (individual living conditions; weight 0.09), in which Chomutov joins the Most, Karviná, and Ostrava-City districts, all with a low index affected mainly by the small share of flats heated by electricity or gas. In Area 7 (health and social care resources; weight 0.10), districts perform above average or average, except for Sokolov, which lags behind in the share of hospital beds (see Table 4).

When using the TOPSIS method with weightings, the group with the highest Health Index is composed of the following districts: Brno-City, Capital city of Praha, Praha-západ, Plzeň-město, České Budějovice, and Praha-východ, and vice versa the districts with the lowest Health Index are Znojmo, Hodonín, Chomutov, Louny, Teplice, Karviná, and Most (see Table 5).

For districts with positive results, the high value of the index in Area 2, education (weight 0.18), and Area 1, economic conditions and social protection (weight 0.19), is clearly prevailing, again with the exception of the district Brno-City. In Area 8, health (weight 0.20), in contrast to the WSA results, there are also some districts with lower rankings and health index values: České Budějovice (especially above-average deaths from respiratory diseases) and Plzeň-City (especially increased spontaneous abortion index and above-average proportion of treated diabetics). In Areas 3, 4, 5, 6, and 7, the districts show a diverse range of rankings and index values. In Area 3 (demographic indicators; weight 0.08), the districts of Praha-východ, Praha-západ, and České Budějovice again rank best, with the district of Plzeň-město on the opposite side. In Area 4 (environmental conditions; weight 0.14), the České Budějovice district positively outperforms all other regions. In Area 5 (individual living conditions; weight 0.09) and Area 6 (road safety and crime; weight 0.04), the districts of Praha-západ and Praha-východ perform positively, but it is clear that they lag behind in Area 7 (health and social care resources; weight 0.10), which reflects the low capacity of medical care, as was also indicated by the WSA method.

For the group of districts with a low Health Index value when calculated using the TOPIS method with weightings, the results were mainly due to the unfavorable situation in Area 1 (economic conditions and social protection; weight 0.19) and Area 2 (Education; weight 0.18). In Area 3 (demographic indicators; weight 0.08), the districts of Western Bohemia have a higher index value, mainly due to the declining of the historically younger age structure, in contrast to the districts of Karviná and Hodonín, which are migration-losing districts with an above-average age index. The situation in Area 4 (environmental conditions; weight 0.14) is negative for most districts, with the exception of Chomutov, which has more favorable values in the annual average concentration of PM_10_ and benzo(a)pyrene. In the case of Area 7 (health and social care resources; weight 0.10), the district of Most is in a good position, especially with regard to an above-average hospital bed capacities index in hospitals and places in social service facilities. There is also a disparity of results within this group of districts in Area 8 (health; weight 0.20): the districts of Znojmo and Hodonín show a positive value of the area index. In Znojmo district, this is due to more favorable results for neonatal mortality and the proportion of births with birth weight below 2500 g, while in Hodonín district, it is due to mortality from diseases of the respiratory system and diabetes mellitus. In both districts, however, the mortality rate due to liver diseases is above average.

## 5. Discussion

The territorial level of districts (LAU 1) was deliberately chosen for the assessment of spatial differentiation of health position inequalities and their factors in the Czech Republic, because international statistics and projects (Eurostat, Euro-Healthy, etc.) in most cases analyze the situation at the national level or at the level of cohesion regions (NUTS2), which in the case of the Czech Republic form regions with heterogeneous geographical, settlement, economic, social, and environmental structures within the regions themselves.

A number of methods are used to assess health position inequalities, among which the creation of indices (composite indicators) is a proven and used method at both the international [53,63,64,65] and national level [66,67,68]. A detailed analysis of the indicators and dimensions of different health indices in previous studies is discussed by [5]. The authors found that the social dimension, including data on education, was found in more than 90% of all studied indexes, the economy dimension was found in 78%, and the policy dimension was found in 56% of all indexes. The indexes were mostly included in 2 up to 6 dimensions with 4 to 76 variables (see [5] for details).

The aim of this article was not only to comprehensively define and assess the spatial differentiation of determinants of health position inequalities, but also to determine the impact of different methods on the final outcomes. In comparing the results according to the WSA and TOPSIS methods, very similar tendencies can be observed:If all eight areas are assigned the same weight of one, then the spatial differentiation of the districts in the Czech Republic is similar using both methods, but the aggregate Health Index in the TOPSIS method achieves a slightly larger range of results, which offers the possibility of using more intervals of the distribution of values.Comparing the benefits when different weightings are given to each area, it is clear that:
(a)The TOPSIS method has a positive benefit for a larger number of districts in terms of a shift in ranking, if the Health Index value increases at the same time;(b)When both methods are used, the Health Index value changes in the districts both positively (increase) and negatively (decrease), but this change in the index is not reflected in a change in the ranking of the best- and worst-performing districts (e.g., Table 2 and Table 3);(c)The allocation of weightings in both methods results in a rearrangement of the ranking of districts for which the Health Index is around the average. The benefits expressed by the change in ranking are not as significant for these districts, in either a positive or a negative sense. At the same time, the change in the index value itself is not nearly as large as for the best- and worst-performing districts (see Figure 1 and Figure 2, the group of districts numbered 2 and 3); and(d)A notable exception, in both methods and when comparing the variant without and with weightings, is the drop in the ranking of the districts of Děčín and Ústí nad Labem (Table 6), to which the district of Český Krumlov is added in the TOPSIS method (Figure 1 and Figure 2, group of districts marked with the number 6).


Demographic and socio-economic indicators are also used in larger national and international comparative studies of health and health status [69,70], etc. Studies illustrate persistent systematic spatial inequalities in health positions at the national level and within EU countries [33]. These spatial inequalities are simultaneously associated with social inequalities at the individual level: the more people are disadvantaged by various components of the living space, the worse their quality of life and health indicators, the more often they become ill, and the lower their life expectancy [62,71]. At the same time, better health usually improves people’s productivity and amplifies the return to education [72], while at lower levels of income, the offspring’s survival probability increases with parental levels of human capital [73].

Health position is a multidimensional concept and the effect of one variable may vary across dimensions. At the same time, the influence of determinants on health position is problematic because these variables correlate with each other to varying degrees, making it sometimes difficult to estimate the net effect of a single variable [74]. Moreover, the strength of the association between health position and its determinants is inconstant across the life course [75]. Nevertheless, indicators are well-established monitoring tools, not only because of their ability to measure, but more specifically because they enable priority setting, policy formulation, and evaluation of these policies [76,77].

## 6. Conclusions

Since 2009, the European Union has made reducing health inequalities one of its priorities, with the support of the Commission Communication Solidarity in Health in the form of the Communication “Reducing Health Inequalities in the European Union” [78]. Having reliable data is subsequently essential for informed decision-making, identifying gaps and better understanding the impacts and consequences of these decisions [79,80,81]. Our analysis brings new information at several levels: (1) in one place, we provide a comprehensive evaluation of population health indicators according to combined data from various databases; (2) we work with data in detailed territorial resolution (districts), which allows targeted measures to reduce health inequalities in local level, neither of these approaches has been applied yet in the Czech Republic and both can be used in other regions or countries; and (3) we present the evaluation of results by various methods as a methodical guide on how to work with data.

The authors are aware of the risk of generalization of the results in the context of analyses using aggregated data. For this reason, ad hoc sociological research is being conducted as a follow-up study to describe health inequalities and possible explanatory factors at the individual level. This research will focus only on the districts that perform worst in terms of health status using aggregated data according to the methods described above. The current situation addresses a number of economic, social, and societal changes that reflect the impact of the COVID-19 epidemic. An effective instrument is needed for decision-making processes at the national and regional level to serve as a tool for selecting appropriate measures to reduce health inequalities. These measures can have an even more significant impact if they are selectively targeted.

## Figures and Tables

**Figure 1 ijerph-18-12275-f001:**
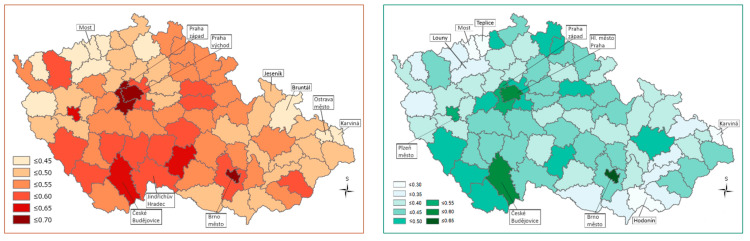
Spatial differentiation of the Health Index values using equal weightings for the different areas—WSA method left side map, TOPSIS right side map ^1^. ^1^ Higher Health Index values are indicated by darker coloring of a district. This also suggests a more favorable situation.

**Figure 2 ijerph-18-12275-f002:**
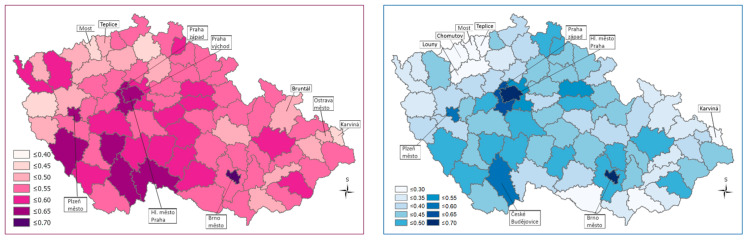
Spatial differentiation based on the calculation of the Health Index values using different weightings—WSA method, left side map, TOPSIS right side map ^1^. ^1^ The higher Health Index values are indicated by the darker color of the districts, suggesting a more favorable situation.

**Figure 3 ijerph-18-12275-f003:**
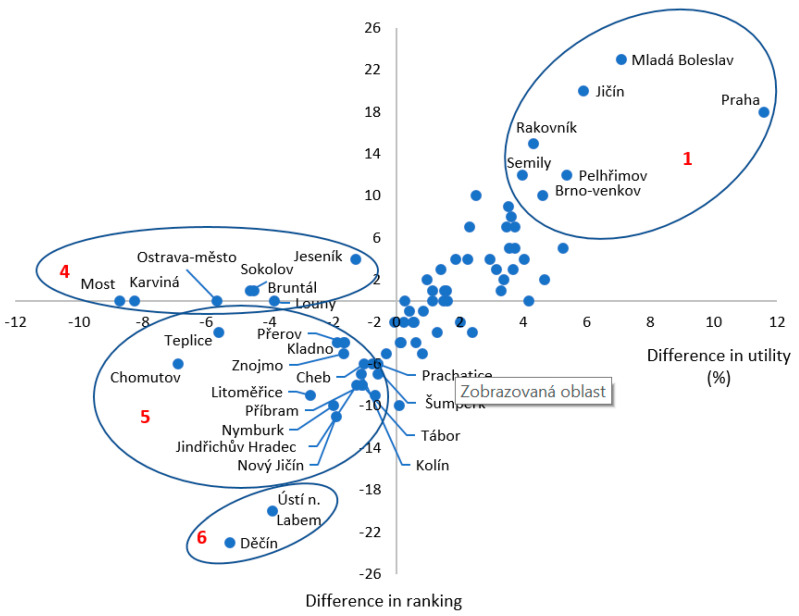
Health Index values calculated using the WSA method.

**Figure 4 ijerph-18-12275-f004:**
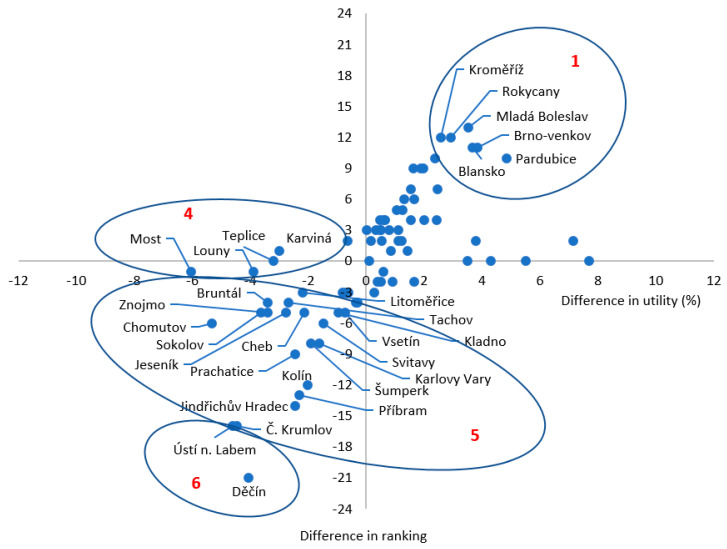
TOPSIS method, difference in ranking, and utility in weight assignment districts of the Czech Republic.

**Table 1 ijerph-18-12275-t001:** Areas assessed in the Health Index and their associated weightings.

Area		Weight
1	Economic conditions and social protection	0.19
2	Education	0.18
3	Demographic indicators	0.08
4	Environmental conditions	0.14
5	Individual living conditions	0.09
6	Road safety and crime	0.04
7	Health and social care resources	0.10
8	Health status	0.20

**Table 2 ijerph-18-12275-t002:** Health Index values calculated using the WSA and TOPSIS method with equal weightings given to the areas (selection of the best and worst five districts of the Czech Republic).

WSA (Weight 1)			TOPSIS (Weight 1)	
Rank	District	Health Index	District	Health Index
1	Praha-západ	0.67	Brno-město	0.63
2	České Budějovice	0.64	Praha Capital City	0.59
3	Brno-město	0.64	České Budějovice	0.56
4	Praha-východ	0.61	Plzeň-město	0.55
5	Jindřichův Hradec	0.60	Praha-západ	0.54
73	Jeseník	0.37	Louny	0.30
74	Bruntál	0.37	Hodonín	0.29
75	Ostrava-město	0.36	Teplice	0.28
76	Most	0.35	Most	0.27
77	Karviná	0.33	Karviná	0.25

**Table 3 ijerph-18-12275-t003:** Health index, WSA, and TOPSIS method, different weightings of areas (selection of the best and worst five districts of the Czech Republic).

	WSA (Different Weightings)		TOPSIS (Different Weightings)	
Rank	District	Health Index	District	Health Index
1	Praha-západ	0.71	Brno-město	0.67
2	Praha-východ	0.66	Hl. město Praha	0.66
3	Brno-město	0.66	Praha-západ	0.61
4	České Budějovice	0.65	Plzeň-město	0.58
5	Hl. město Praha	0.65	České Budějovice	0.57
73	Bruntál	0.32	Chomutov	0.27
74	Chomutov	0.32	Louny	0.26
75	Ostrava-město	0.31	Teplice	0.24
76	Most	0.26	Karviná	0.22
77	Karviná	0.24	Most	0.21

**Table 4 ijerph-18-12275-t004:** WSA weighted.

			Determinants of Health														Health Indicators	
			Area 1		Area 2		Area 3		Area 4		Area 5		Area 6		Area 7		Area 8	
	Health Index		Economic Conditions and Social Protection		Education		Demographic Indicators		Environmental Conditions		Individual Living Conditions		Road Safety and Crime		Health and Social Care Resources		Health Status	
District	Rank	Index Value	Rank	Index Value	Rank	Index Value	Rank	Index Value	Rank	Index Value	Rank	Index Value	Rank	Index Value	Rank	Index Value	Rank	Index Value
Praha-západ	1	0.710	9	0.698	3	0.832	8	0.582	46	0.476	1	0.728	14	0.791	75	0.234	4	0.757
Praha-východ	2	0.657	1	0.803	4	0.697	4	0.593	71	0.195	3	0.656	27	0.767	72	0.294	9	0.723
Brno-město	3	0.656	66	0.437	2	0.912	12	0.573	47	0.476	54	0.381	44	0.713	3	0.800	13	0.695
České Budějovice	4	0.650	27	0.627	7	0.569	11	0.577	14	0.740	32	0.474	36	0.747	9	0.582	29	0.622
Cap. C. Praha	5	0.650	7	0.721	1	0.999	55	0.463	75	0.181	48	0.399	75	0.514	28	0.467	3	0.762
Jeseník	69	0.361	68	0.416	67	0.173	74	0.388	19	0.710	25	0.491	57	0.651	32	0.462	68	0.438
Louny	70	0.345	70	0.331	68	0.161	15	0.549	41	0.558	15	0.529	74	0.519	20	0.504	74	0.359
Sokolov	71	0.331	67	0.426	77	0.000	6	0.614	9	0.807	44	0.412	70	0.576	70	0.313	73	0.361
Teplice	72	0.327	52	0.533	70	0.152	28	0.522	66	0.244	24	0.498	53	0.673	11	0.559	77	0.307
Bruntál	73	0.324	74	0.327	69	0.161	19	0.539	21	0.707	39	0.429	66	0.607	39	0.432	72	0.375
Chomutov	74	0.320	73	0.331	75	0.101	7	0.614	20	0.709	58	0.364	64	0.608	10	0.565	76	0.315
Ostrava-město	75	0.306	76	0.331	10	0.469	1	0.661	77	0.014	75	0.266	63	0.613	6	0.634	71	0.382
Most	76	0.258	75	0.325	72	0.128	3	0.641	68	0.212	74	0.269	48	0.698	7	0.619	75	0.321
Karviná	77	0.245	77	0.217	59	0.242	10	0.581	76	0.017	56	0.374	24	0.771	33	0.460	69	0.430

**Table 5 ijerph-18-12275-t005:** TOPSIS weighted.

			Determinants of Health														Health Indicators	
			Area 1		Area 2		Area 3		Area 4		Area 5		Area 6		Area 7		Area 8	
	Health Index		Economic Conditions and Social Protection		Education		Demographic Indicators		Environmental Conditions		Individual Living Conditions		Road Safety and Crime		Health and Social Care Resources		Health Status	
District	Rank	Index Value	Rank	Index Value	Rank	Index Value	Rank	Index Value	Rank	Index Value	Rank	Index Value	Rank	Index Value	Rank	Index Value	Rank	Index Value
Brno-město	1	0.673	66	0.453	2	0.867	53	0.368	50	0.378	54	0.425	40	0.687	1	0.877	20	0.598
Cap. C. Praha	2	0.663	7	0.640	1	0.999	38	0.410	75	0.176	53	0.427	73	0.511	6	0.640	5	0.676
Praha-západ	3	0.612	22	0.593	3	0.828	1	0.704	49	0.379	32	0.528	6	0.785	75	0.144	2	0.701
Plzeň-město	4	0.584	10	0.621	5	0.670	72	0.313	59	0.295	68	0.336	58	0.597	2	0.873	42	0.548
České Budějovice	5	0.574	30	0.572	7	0.583	11	0.499	17	0.532	42	0.473	43	0.679	8	0.560	31	0.568
Praha-východ	6	0.546	2	0.686	4	0.709	2	0.704	71	0.184	33	0.520	14	0.753	73	0.198	3	0.692
Znojmo	69	0.295	73	0.38357	71	0.149	35	0.418	39	0.450	39	0.509	33	0.711	39	0.363	17	0.604
Hodonín	70	0.290	59	0.47157	62	0.238	71	0.323	57	0.300	64	0.351	12	0.757	50	0.339	21	0.594
Chomutov	73	0.276	74	0.369	75	0.114	8	0.524	18	0.528	66	0.348	61	0.587	25	0.402	76	0.367
Louny	74	0.262	69	0.420	68	0.172	17	0.473	48	0.402	29	0.538	75	0.471	53	0.331	73	0.400
Teplice	75	0.244	46	0.535	70	0.168	34	0.419	66	0.251	44	0.465	57	0.615	26	0.395	77	0.353
Karviná	76	0.223	77	0.311	62	0.249	55	0.360	76	0.049	47	0.462	18	0.748	35	0.374	69	0.433
Most	77	0.214	71	0.535	74	0.129	13	0.486	68	0.201	73	0.318	35	0.648	14	0.473	72	0.402

**Table 6 ijerph-18-12275-t006:** Comparison of changes in Health Index values and ranking of the districts of Děčín and Ústí nad Labem.

	WSA without Weightings		WSA Weighted		TOPIS without Weightings		TOPIS Weighted	
District	Index Value	Rank	Index Value	Rank	Index Value	Rank	Index Value	Rank
Děčín	0.50	36	0.44	59	0.44	22	0.40	43
Ústí nad Labem	0.49	38	0.46	58	0.44	15	0.42	31

## Data Availability

Publicly available datasets were analyzed in this study. The data come from publicly available databases of the Czech Statistical Office, the Institute of Health Information and Statistics of the Czech Republic, the Ministry of Labor and Social Affairs of the Czech Republic, the Czech Hydrometeorological Institute, and the Czech Household Panel Survey. The data are from 2016, 2017, 2018, and 2019, the only indicators in relation to the level of education and housing conditions come from the 2011 Census of Population, Houses. and Flats.

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
