# Peer review of "Inequalities in Health: Methodological Approaches to Spatial Differentiation"

_ijerph, 2021, doi:10.3390/ijerph182312275_

Round 1
Reviewer 1 Report
This paper is very interesting and well prepared. The use of the discussed indicator is interesting.
I suggest only to improve quality of the figures 1 and 2.
Good luck!
Author Response
Dear reviewer, thank you for your comments. The quality of figure 1 and 2 will be improved.
Best regards.
Reviewer 2 Report
Referee report on "Inequalities in health: methodological approaches to spatial differentiation"
This paper’s major contribution is to provide a composite indicator (health index) in eight elements for the assessment of inequalities in health in Czech Republic. The paper is interesting and easy to follow. My comments are relatively minor.
- Please add some more information on your own contribution in the Introduction.
- Page 2, Line 84: Decipher the term CEE. It is not well-known for everyone.
- In the same paragraph, you compare people with similar levels of educations across countries that are very different economically. It would be of interest to compare more educated people in the CEE region to less educated people in richer and healthier regions in Europe. Is there a threshold level of education in the CEE countries, from which more educated people in the CEE overcome the lower educated people in Western Europe?
- Table 1 on page 4: Explain the reason for choosing these particular weights. This is the major point of your study and it should be introduce and explained well.
- In the last paragraph on this page you present the regions you’ve compared. You speak about it later, but maybe you can give us some feelings already here can these two groups be classified as wealthy versus poorer regions? A couple of sentences with economic figures will be enough.
- Is it possible to provide any data on the presence of ethno/ racial minorities (especially, Roma and recent immigrants from outside Europe) in these regions?
- When you speak about socio-economic indicators in the paragraph after Table 6 in the Discussion (lines 397 – 403), I would suggest making the point clearer by adding the following sentence:
“At the same time, better health usually improves people’s productivity and amplifies the return to education (Azarnert, 2020), while at lower levels of income the offspring’s survival probability increases with parental levels of human capital (Azarnert, 2006).”
- Re-write Conclusion. Some of the general ideas addressed in Conclusion should be better relegated to the Introduction. In Conclusion you should better concentrate on the presentation of the particular contributions of the present paper.
- Line 460: Health Aff -> Health Affairs
References
Azarnert LV (2006) ‘Child Mortality, Fertility and Human Capital Accumulation’ Journal of Population Economics, 19(2): 285–297
Azarnert LV (2020) ‘Health Capital Provision and Human Capital Accumulation’ Oxford Economic Papers, 72(3): 633–650
Author Response
Please add some more information on your own contribution in the Introduction. Added
Page 2, Line 84: Decipher the term CEE. It is not well-known for everyone. Explained
In the same paragraph, you compare people with similar levels of educations across countries that are very different economically. It would be of interest to compare more educated people in the CEE region to less educated people in richer and healthier regions in Europe. Is there a threshold level of education in the CEE countries, from which more educated people in the CEE overcome the lower educated people in Western Europe? Added on the example of life expectancy as a comprehensive indicator of mortality rates.
Table 1 on page 4: Explain the reason for choosing these particular weights. This is the major point of your study and it should be introduce and explained well. Explained
In the last paragraph on this page you present the regions you’ve compared. You speak about it later, but maybe you can give us some feelings already here can these two groups be classified as wealthy versus poorer regions? A couple of sentences with economic figures will be enough. Explained
Is it possible to provide any data on the presence of ethno/ racial minorities (especially, Roma and recent immigrants from outside Europe) in these regions? Explained
When you speak about socio-economic indicators in the paragraph after Table 6 in the Discussion (lines 397 – 403), I would suggest making the point clearer by adding the following sentence: “At the same time, better health usually improves people’s productivity and amplifies the return to education (Azarnert, 2020), while at lower levels of income the offspring’s survival probability increases with parental levels of human capital (Azarnert, 2006).” Added
Re-write Conclusion. Some of the general ideas addressed in Conclusion should be better relegated to the Introduction. In Conclusion you should better concentrate on the presentation of the particular contributions of the present paper. Text edited
Line 460: Health Aff -> Health Affairs - Corrected
References - Added
Reviewer 3 Report
This is a paper on spatial differentiation of health inequalities although it has to be said that much of the focus is on the use of two different methods of weighting in relation to an index that has been created. With this in mind, I wonder if this is the most appropriate journal for it.
The opening statement in the abstract is somewhat misleading and it appears that the authors are perhaps not well versed in the broader literature around inequalities and health. Indeed this is made more apparent when the authors state in the conclusion that in order to understand why some populations or groups are healthier than others, the monitoring of determinants should go beyond measuring health outcomes. However, there is a whole public health literature which does exactly this, which isn’t referred to in the paper. In terms of language, the meaning is sometimes quite difficult to grasp. The paper would benefit from being reviewed by someone with English as a first language
The authors suggest that the relationship between health determinants and health outcomes is confirmed by a recent study in the US. Something as complex as health determinants and health outcomes cannot be determined by one study, but see comment above about the wealth of literature that isn’t referred to (P2 line 69).
Provide the name of the author where you are referring to them directly, not just a ref number (p2 line 92)
The methodology of creation of the health index is not described clearly which makes it difficult for the reader to get a sense of how valid the approach might have been. It isn’t clear how the range of the original 143 available indicators were chosen and by whom. There doesn’t seem to be any theoretical approach used here beyond including any available indicator that was available in the data bases and appeared relevant. The inclusion of health as an area seems a bit unusual. Is the model not attempting to demonstrate how social determinants impact on health. However, health is also presented (conceptually at least) as an outcome of the indicators and areas being considered. Further, the weighting of each area in the index was assessed using Delphi with experts. This seems most unusual to assign weightings based on opinion. (P3)
Indicators were dropped when correlation was 0.8 (or -0.8) or higher. This means that there was no doubt strong and significant correlation between many of the indicators included.
The authors subsequently point out this out about the correlation between factors but many of these are already recognized and well known I’m not knowledgeable enough about stats to comment further on this. level of collinearity between indicators, but it seems reasonable to conclude that it would have had a significant impact this might have had on the findings.
As a mixed methods public health researcher, I can’t comment on the statistical approaches used (Weighted sum approach and the TOPSIS method) as my knowledge isn’t sufficient However I note that the results seem to focus more on the impact of altering the weightings as opposed to the validity and reliability of the methods used. Should there not have been some attempt to validate the findings? For example 4.3 Is the main contributor to the scores a product of the original weighting decisions or can you be certain that it is an objective finding? (line 266 p8)
Are you clear about what the index is telling us that we don’t already know? You seem to have modelled that areas with higher socio-economic indicators have better health and vice versa but this is a well known spatial correlation which would no doubt hold out in many different international settings.
You mention in the conclusion that in order to understand why some populations or groups are healthier than others, the monitoring of determinants should go beyond measuring health outcomes – but there is a whole literature that discusses this.
Overall, I have concerns about the validity of the methodology and the findings and the fact that much of the literature in the field has not been acknowledged. However, in terms of methodology, I think the view of a statistician or a more stats based epidemiologist would be most useful to take into account here.
Round 2
Reviewer 3 Report
I don't feel the reviewers responses to my points are very robust.
For example I give two responses below ;
Response two
They argue that “the purpose of the article is not to present a comprehensive review of the literature on health inequalities, but above all an overview of selected major health factors (eg education), information on methods and practical and applied methodological approaches.”
This doesn’t really answer the point raised that there is a wealth of literature that isn’t referred to (P2 line 69). It also ignores the point I raised that, referring to one study with regards to something as complex as the relationship between health determinants and health outcomes is not best practice. Further, although of course the purpose is not to present a comprehensive review of the literature, this is the starting point of the research and an entire body of public health evidence is absent. The authors have made no attempt to address this.
Response 7: The authors state that a value of 0.8 is the default value for correlations. We reflected the resulting correlations; indicators that exceeded the set limit of 0.8 were excluded from the data set.
I find this odd. 0.8 is not a default value. Plenty of authors would set it lower. This isn't justified wither in the responses and neither do the authors discuss the potential impact of collinearity.
These are just two examples. Given that the other reviewers have not commented on the methods, I really do recommend an epidemiologist with strong stats expertise also reviews this paper as I suspect reviewers have been selected on their expertise in inequalities rather than the methods used including myself.
